# Artificial Intelligence-Enhanced Quantitative Ultrasound for Breast Cancer: Pilot Study on Quantitative Parameters and Biopsy Outcomes

**DOI:** 10.3390/diagnostics14040419

**Published:** 2024-02-14

**Authors:** Hyuksool Kwon, Seokhwan Oh, Myeong-Gee Kim, Youngmin Kim, Guil Jung, Hyeon-Jik Lee, Sang-Yun Kim, Hyeon-Min Bae

**Affiliations:** 1Laboratory of Quantitative Ultrasound Imaging, Seoul National University Bundang Hospital, Seong-nam 13620, Republic of Korea; jinuking3g@snubh.org (H.K.); joseph9337@kaist.ac.kr (S.O.); 2Imaging Division, Department of Emergency Medicine, Seoul National University Bundang Hospital, Seong-nam 13620, Republic of Korea; 3Electrical Engineering Department, Korea Advanced Institute of Science and Technology, Daejeon 34141, Republic of Korea; mgkim@barreleye.co.kr (M.-G.K.); youngmin2007@kaist.ac.kr (Y.K.); jgl97123@kaist.ac.kr (G.J.); dlguswlr0811@kaist.ac.kr (H.-J.L.); kmjmksy@kaist.ac.kr (S.-Y.K.)

**Keywords:** breast cancer, quantitative ultrasound, artificial intelligence

## Abstract

Traditional B-mode ultrasound has difficulties distinguishing benign from malignant breast lesions. It appears that Quantitative Ultrasound (QUS) may offer advantages. We examined the QUS imaging system’s potential, utilizing parameters like Attenuation Coefficient (AC), Speed of Sound (SoS), Effective Scatterer Diameter (ESD), and Effective Scatterer Concentration (ESC) to enhance diagnostic accuracy. B-mode images and radiofrequency signals were gathered from breast lesions. These parameters were processed and analyzed by a QUS system trained on a simulated acoustic dataset and equipped with an encoder-decoder structure. Fifty-seven patients were enrolled over six months. Biopsies served as the diagnostic ground truth. AC, SoS, and ESD showed significant differences between benign and malignant lesions (*p* < 0.05), but ESC did not. A logistic regression model was developed, demonstrating an area under the receiver operating characteristic curve of 0.90 (95% CI: 0.78, 0.96) for distinguishing between benign and malignant lesions. In conclusion, the QUS system shows promise in enhancing diagnostic accuracy by leveraging AC, SoS, and ESD. Further studies are needed to validate these findings and optimize the system for clinical use.

## 1. Introduction

Breast cancer ranks as the second most common cancer worldwide and remains the leading cause of cancer death among women [1]. Early detection plays a pivotal role in enabling effective treatment and enhancing survival rates. Mammography serves as the gold standard for breast cancer screening; however, its sensitivity decreases in women with dense breast tissue, a condition more prevalent in younger women [2]. Ultrasonography has gained recognition as a complementary tool to mammography for breast cancer screening, especially in women with dense breasts or those with an elevated risk for breast cancer.

Quantitative ultrasonography (QUS) is an advanced imaging technique that facilitates the quantification of multiple tissue properties through sound waves [3]. Unlike traditional ultrasound imaging, which offers qualitative visual information about the tissue, QUS generates numerical data, enabling more objective and precise evaluations. QUS has proven valuable in assessing diverse medical conditions, including breast cancer [4,5]. By supplying quantitative measurements of tissue characteristics, QUS holds the potential to augment the accuracy of breast cancer detection, characterization, and monitoring and to deepen our comprehension of the underlying mechanisms driving breast cancer development and progression.

Despite the potential advantages of QUS in breast cancer detection and management, its implementation has encountered several obstacles. One major limitation of QUS is the variability in its sensitivity and specificity, which depends on the specific QUS method employed and the population being examined. Moreover, the absence of standardized protocols for QUS acquisition and analysis, along with variability in interpretation, can result in inconsistencies in QUS outcomes. Additionally, the cost-effectiveness of QUS in comparison to other imaging modalities remains uncertain. These factors have impeded the routine clinical use of QUS for breast cancer [6]. To address these challenges, we have developed a single-probe ultrasonic imaging system capable of capturing multi-variable QUS profiles. This system employs artificial intelligence (AI) to perform tomographic reconstruction of attenuation coefficient (AC), speed of sound (SoS), effective scatterer diameter (ESD), and effective scatterer concentration (ESC) [7,8,9] of breast ultrasound radiofrequency (RF) signals for malignancy diagnosis, in real-time using a single scalable neural network.

The objective of this pilot study is to validate the efficacy of our AI-based QUS algorithm, which reconstructs AC, SoS, ESD, and ESC of breast ultrasound RF signals. We aim to analyze these parameters and juxtapose the results with biopsy outcomes to underscore the utility of the proposed algorithm in breast cancer diagnosis.

## 2. Materials and Methods

The study was divided into two phases: Phase 1, which created an algorithm with QUS parameters to determine and analyze QUS parameters, and Phase 2, which used the completed algorithm to observe patients with real breast masses.

### 2.1. Phase 1. Development of Algorithm with New Quantitative Ultrasound Parameters for Diagnosis of Breast Cancer

Tissue biomechanical characteristics affecting ultrasound wave propagation are divided into two categories, which are acoustic and structural properties (Figure 1). Tissue AC, SoS, and density are considered acoustic properties that determine how the tissue interacts with ultrasound waves. The AC is the degree to which ultrasound waves are attenuated in tissue. Hence, the tissue AC is related to the amplitude of the RF signal. The SoS influences the propagation speed of ultrasound waves. The tissue speed of sound determines the arrival time of the pulse-echo signal and is reflected in the phase of the received ultrasound signal. The tissue density formulates acoustic impedance in conjunction with sound speed. The acoustic properties regulate the reflection and refraction of an ultrasound pulse in heterogeneous tissue. The scattering of ultrasonic waves (ESC and ESD) occurs due to the heterogeneity of speed of sound and density in tissue, and the size and concentration of the scattering body that determines the scattering of ultrasonic waves are considered structural properties. The geometry of micro-tissue components governs the ultrasound scattering effect, and such tissue structural properties are recognized by analyzing envelop statistics of the received ultrasound signal. The proposed quantitative imaging system is configured to reconstruct the tissue AC, SoS, ESD, and ESC of the tissue by analyzing amplitude, phase, and envelop statistics of the RF signal, respectively. These parameters are estimated through methods that analyze the frequency domain characteristics of the ultrasound signal.

#### 2.1.1. Quantitative Ultrasound Parameters

(1) Attenuation Coefficient (AC)

The AC quantifies the loss of acoustic energy during the propagation of ultrasound waves. The wave absorption, reflection, refraction, and scattering determine the magnitude of acoustic attenuation. The acoustic attenuation coefficient is described using the unit of decibel per centimeter per megahertz (dB/cm/MHz). The AC parameter is formulated by analyzing the amplitude decay of the received RF signal. 

(2) Speed of Sound (SoS)

The SoS refers to the ultrasound propagation speed. The tissue SoS is determined by the square root of bulk elasticity modulus divided by the tissue density. The speed of sound is expressed in meters per second (m/s). Heterogeneous tissue SoS distribution affects the arrival time of the reflected ultrasound signal and results in the phase aberration of each plane-wave signal. The proposed system extracts tissue SoS by analyzing the phase difference of each steering angle plane wave. 

(3) Effective Scatterer Diameter (ESD)

The tissue ESD quantifies the diameter of the tissue microstructures. The ESD is expressed in the unit of micro-meter (um). The geometry of the tissue microstructures affects the aspects of the ultrasound scattering effect and is mathematically described through the Gaussian form factor [10,11]. The system obtained the ESD feature by interpreting the backscatter distribution of the measured RF signal. 

(4) Effective Scatterer Concentration (ESC)

The ESC refers to the number of scattering sources in the unit area. The unit of ESC is expressed as the scatterer numbers per square of the wavelength (#/wavelength^2). The ESC determines the magnitude of the ultrasound backscattering effect and is calculated by analyzing the scatterer probability distribution of the RF signal corresponding to the unit area.

#### 2.1.2. Development of Quantitative Imaging Algorithm

The learning-based quantitative imaging algorithm was made to reconstruct multi-variable quantitative images from the RF signals (Figure 2). The algorithm is implemented based on the encoder-decoder structure, where the encoder extracts tissue property relevant features **q** from the RF signal, and the decoder generates a QUS image using **q**. A conditional feature extraction scheme is employed to adaptively re-parametrize the neural network with the corresponding reconstruction objective (Figure 3). Namely, when AC is given as a condition parameter, the algorithm extracts AC-relevant quantitative features from the RF and generates the AC images as the output. Through the conditional feature parameterization scheme, the algorithm provides *N* (=4) variable QUS images with a single learned representation. 

The acoustic dataset is generated using k-Wave [12] simulation toolbox in MATLAB (R2022b) as proposed in [13,14]. The numerical simulation is set to cover general human soft-tissue biomechanical features and is used to train the algorithm. The algorithm employs RF signals, and condition variables as inputs of the system. The conditional variable is composed of a one-hot vector that determines the reconstruction objective of the system. For example, [1000] indicates the attenuation coefficient parameter. If [1000] is employed as neural network input, the attenuation coefficient image is reconstructed as an output of the neural network. The neural network is implemented on the basis of an encoder-decoder structure. The encoder is composed of a two-dimensional convolutional operation, conditional instance normalization [15], a non-linear activation function, and a down-sampling layer. The conditional instance normalization employs condition parameters to reconfigure the mean $\beta_{cond}$ and standard deviation $\gamma_{cond}$ of the received signal. The conditional instance normalization is formulated as follows,
$$Conditional\;Instance\;Normlization\left(x,cond\right)=\gamma_{cond}\frac{x-{µ}_{\;}\left(x\right)}{\sigma_{\;}\left(x\right)}+\beta_{cond}$$

Through the conditional instance normalization scheme, distinct mean and standard deviation feature is trained to adaptively extract quantitative features relevant to the condition variable. Consequently, the encoder module extracts compressed quantitative features ($q\;\sim R^{16\times 16\times 512}$), which contains tissue biomechanical features of a selected quantitative parameter. 

The decoder generates a high-resolution quantitative image using q. For detailed image reconstruction, the decoder is composed of a parallel multi-resolution subnetwork [16]. Each subnetwork is responsible for generating the corresponding resolution ($\in R^{16\times 16},\;R^{32\times 32},\;R^{64\times 64},\;and\;R^{128\times 128}$)) quantitative image. The subnetwork is formulated with a residual convolutional layer [17] for ease of network optimization. Outputs of each subnetwork are up-sampled and concatenated into the highest resolution feature representation and reconstructed the resolution-enhanced network outcome through bottleneck convolution.

The optimization procedure of the proposed quantitative imaging neural network is introduced in Table 1. The 16.5 k datasets are split into the train (*n* = 15.5 k) and test (*n* = 1.0 k) datasets. Each data is composed of an *RF* signal, condition variable (*cond*), and ground truth quantitative image (*I_cond_*). A batch of size 8 is configured for the training of the neural network. The learning objective of the system is described as follows:$$\theta^{*}=arg\underset{\theta }{\min\;}E_{\mathrm{Ɗataset}}\left[{\left\|I_{\mathrm{cond}}-\theta \left(\mathrm{RF},\mathrm{cond}\right)\right\|}^{2}\right]+L_{R}+L_{2},\;\;$$
$$where,\;L_{R}=\sum_{R}\left[{\left\|y_{R}-\theta_{R}\left(RF,\;cond\right)\right\|}^{2}\right]$$
$$and,\;{\;L}_{2}=\sum_{i}w_{i}^{2}\;$$

The neural network tries to minimize the squared difference between the neural network output $\theta \left(RF,\;cond\right)$ and corresponding ground truth $I_{cond}$. $L_{R}\;$ is employed to parameterize each resolution subnetwork. The $L_{2}$ regularized the weight $w_{i}^{\;}$ of convolutional layers and prevents the neural network from over-fitting.

Breast tissue mimicking acoustic phantoms are modeled to simulate ultrasonic wave propagation. The anatomy of the phantom is generated using a number of MRI-driven realistic breast tissue segments (UWCEM breast phantom repository [18]). The shape of the breast tumor is modeled using breast B-mode images of 600 female subjects with malignant (*N* = 210) and benign (*N* = 445) lesions [19]. The wave propagation property is set to cover general soft-tissue biomechanical characteristics. The probe is placed along the skin layer, and the region of interest of size 50 × 50 mm is configured correspondingly. The ultrasound simulation is implemented using the k-Wave simulation toolbox in MATLAB and imitates ultrasound measurement of 128 elements with a 5 MHz linear probe. Through the simulation, an RF signal of each 128-probe element is obtained and configures the dataset for the training of the neural network.

The choice to use a 5 MHz transducer for the study, despite breast ultrasound typically being performed with higher frequency transducers (8–9 MHz), can be attributed to several factors. Lower frequency transducers, like the 5 MHz, have deeper penetration capabilities, which can be crucial for visualizing deeper breast tissues and lesions. This can be especially beneficial in patients with denser breast tissues where higher frequencies may not penetrate as effectively. Moreover, the 5 MHz transducer can still provide sufficient resolution for the quantitative analysis required in this study while also allowing for a broader range of tissue characterization due to its wavelength properties. These characteristics make the 5 MHz transducer a suitable choice for the specific requirements and objectives of the pilot study on AI-enhanced quantitative ultrasound for breast cancer diagnosis.

#### 2.1.3. Implementation Details

The proposed quantitative imaging system is accelerated with Geforce RTX 3090 Ti (Nvidia, Santa Clara, CA, USA) graphics processing unit. The neural network was optimized with the stochastic gradient descent method using Adam [20] with a learning rate $10^{-4}$ ($\beta_{1}$ = 0.9$\;\beta_{2}$ = 0.999). The neural network is trained up to 120 epochs, which is determined based on the convergence of validation dataset assessed mean normalized absolute error (MNAE), peak signal-to-noise ratio (PSNR), and structural similarity index (SSIM) metrics. Dropout [19] with a retention probability of 0.5 is implemented for the generalization of the neural network. A figure illustrating the training and validation loss as a function of training epochs is added as a Appendix A. 

#### 2.1.4. Quantitative Ultrasound Measurement System

A Vantage 64LE (Verasonics, Inc., Kirkland, WA, USA) ultrasound system featuring 128 elements and a 5 MHz center frequency linear array probe (5LE, Humanscan, Inc., Siheung, Republic of Korea) is employed to measure ultrasound signal, $RF\subset R^{128\times 3018}$. A plane-wave ultrasound signal is a transmission beamformed with steering angle θ, by formulating pulse emission delay of each ultrasound element as:$$T_{delay\;}\left(\theta ,\;n_{element}\right)=\frac{n_{element}d_{pitch\;}\sin\left(\theta \right)}{c_{tissue}\;}$$

The $d_{pitch\;}$ and $n_{element\;}$ denote pitch size and order of the probe element, respectively. The $c_{ttisue\;}$(=1540 m/s) is the average speed of the sound of tissue. The ultrasound RF signal is received at a 62.5 MHz sampling rate and is employed to extract quantitative features of the measured tissue. 

#### 2.1.5. QUS Formula Using AC, SoS, ESD, and ESC

The model’s results were presented in terms of odds ratios, standard errors, z-scores, *p*-values, and 95% confidence intervals for AC, SoS, ESD, and ESC. The calculated log (odds) values from the logistic regression model were then transformed into the QUS formula using the sigmoid function [2].

### 2.2. Phase 2: Using Completed Algorithm to Observe Patients with Real Breast Masses

#### 2.2.1. Patient Enrollment

The enrollment of patients conducted over a six-month period from September 2022 to March 2023 received approval from the Institutional Review Board at Seoul National University Bundang Hospital (IRB number: B-1910-570-301) and was registered on ClinicalTrials.gov (identifier: NCT05836246). All procedures performed in this study were in strict accordance with the relevant guidelines and regulations as established by the respective overseeing institutions and ethics committees. 

Participants were selected using a consecutive sampling method, encompassing individuals who had breast masses identified through ultrasound examinations and were scheduled for a biopsy. Those who had undergone surgery or anti-cancer treatments for breast conditions were excluded to ensure that the QUS parameters were measured from undisturbed tissue characteristics. This selection strategy aimed to maintain the integrity of the study’s findings and provide results generalizable to a broader patient population. Informed consent was obtained from all individual participants included in the study.

#### 2.2.2. Imaging and Data Acquisition

In this study, ultrasound imaging techniques were carried out in strict accordance with standard clinical protocols, guaranteeing the trustworthiness and consistency of the examinations. These protocols encompassed a multitude of stages, including patient preparation prior to the ultrasound examination, proper adjustment of ultrasound device parameters, optimal patient positioning, and execution of the ultrasound scan. Data acquisition was undertaken by two seasoned radiologists, each with over 20 years of experience and faculty positions at a university hospital, ensuring a high level of expertise and accuracy in data collection. Upon the acquisition of B-mode ultrasound data, the QUS algorithm was applied to the collected RF data to estimate the AC, SoS, ESD, and ESC parameters of each participant’s breast mass. Following this, a biopsy was performed on the lesion. Diagnostic reports of biopsy specimens served as the ground truth to accurately classify the type of tumor, thus enabling a thorough comparison of the QUS results with the histopathological findings.

#### 2.2.3. Image Evaluation by Radiologist Readers

Two board-certified radiologists, each possessing over 20 years of professional experience in breast ultrasound, were recruited as independent readers for the image evaluation. Both radiologists were blinded to their involvement in the study, patients’ clinical information, biopsy results, and QUS findings. The B-mode ultrasound images were randomly presented to the radiologists. For each patient, the radiologists independently assessed the B-mode ultrasound images and provided Breast Imaging-Reporting and Data System (BI-RADS) categories. The radiologists also provided comments on the image quality, artifacts, and any difficulties encountered during the evaluation.

#### 2.2.4. Image Confirmation by Biopsy

To validate the findings from the QUS and B-mode ultrasound images, biopsy outcomes were used as the reference standard to confirm the presence or absence of malignant lesions in the breast tissue. Patients with suspicious findings on B-mode ultrasound images underwent a biopsy procedure to obtain tissue samples for histopathological analysis. The biopsies were performed using either ultrasound-guided core needle biopsy or fine needle aspiration techniques, as deemed appropriate by the radiologist. Fine Needle Aspiration Biopsy plays a critical role in breast cancer diagnosis as it involves the extraction of cells from breast lesions for microscopic examination. FNAB serves as a minimally invasive method to obtain tissue samples, which are then analyzed to differentiate between various pathologies, including Invasive Ductal Carcinoma (IDC), Ductal Carcinoma In Situ (DCIS), and benign lesions. The pathology report generated from FNAB is considered the ground truth because it provides a definitive diagnosis. The biopsy samples were then sent for histopathological evaluation by a board-certified pathologist. The pathologist evaluated the biopsy samples for the presence of malignant cells, as well as the histological type and grade of the tumor, if present. The results of the histopathological analysis were used as the reference standard to classify the lesions as benign or malignant.

### 2.3. Outcomes

#### 2.3.1. Primary Outcome

The main objective of this study was to determine the ability of QUS imaging parameters, including AC, SoS, ESD, and ESC, to differentiate between benign and malignant breast lesions. 

#### 2.3.2. Secondary Outcomes

The secondary objectives of this study included evaluating the correlation between QUS parameters and Breast Imaging-Reporting and Data System (BI-RADS) categories, assessing the performance of a logistic regression model incorporating QUS parameters in predicting benign and malignant breast tumors and investigating the differences in QUS parameters between Invasive Ductal Carcinoma (IDC) and Ductal Carcinoma In Situ (DCIS).

#### 2.3.3. Statistical Analysis

The classification model was evaluated by quantifying the subject’s breast mass using our QUS AI model and comparing it with biopsy results. Data are expressed either as mean ± standard deviation [median; interquartile range] for continuous data or median (range) for categorical data. Nonparametric analysis (Mann–Whitney U test) was utilized for continuous data that did not follow a normal distribution. The rank sum test was also used to compare paired data and to compare grades. In the logistic regression analysis, we utilized independent variables AC, SoS, ESD, and ESC to predict the binary outcome of benign and malignant tumors. The model’s results were presented in terms of odds ratios, standard errors, z-scores, *p*-values, and 95% confidence intervals for each independent variable. The calculated log (odds) values from the logistic regression model were then transformed into probabilities using the sigmoid function [21]. This conversion allowed for the determination of the likelihood of a tumor being benign or malignant based on the specified threshold value. Receiver operating characteristic curves (ROC) and the respective areas under the ROC (AUC) were calculated. The 95% confidence interval (CI) for the AUC was estimated by bootstrapping, and the CIs for metrics were calculated with the use of the Clopper-Pearson test. The analyses were performed with Stata software (version 17.0; Stata). A two-sided *p*-value less than 0.05 was considered to indicate a statistically significant difference.

## 3. Results

### 3.1. QUS Formula Using AC, SoS, ESD, and ESC

Using the logistic regression analysis (Table 2) with independent variables AC, SoS, ESD, and ESC, we predict benign or malignant tumor outcomes. The formula is:$$\log\left(odds\right)=-31.7558+10.6396\;*\;AC+0.0210\;*\;SOS-0.0975\;*\;ESD+0.0805\;*\;ESC\;$$

We can apply the sigmoid function [sigmoid(x) = 1/(1 + exp(−x))] to the obtained log(*odds*) value to estimate the probability of a tumor being benign or malignant, which aids in distinguishing between the two outcomes.

### 3.2. Using Completed Algorithm to Observe Patients with Real Breast Masses

#### 3.2.1. Participant Characteristics

During the six-month study period, ultrasound B-mode images and QUS were obtained from 55 patients. A comparison of benign and malignant lesions among study participants is presented based on age, tumor size, BI-RADS categories, and pathology outcomes (Table 3). The mean age for benign lesions is 41.9 (±1.8), while the mean age for malignant lesions is 53.8 (±2.3). The median tumor size is 14 mm (IQR 10–17 mm) for benign lesions and 20.5 mm (IQR 14–28 mm) for malignant lesions. Regarding the BI-RADS categories, all lesions in categories 1 and 2 (3% and 14% of the total, respectively) are benign. Category 3 has a total of 6 lesions (17% of the total), with 5 benign and 1 malignant lesion. Category 4 has a total of 3 lesions (8% of the total), with 2 benign and 1 malignant lesion. Categories 5 and 6 have 1 (3% of the total) and 20 lesions (64% of the total), respectively, with all lesions in category 6 being malignant. Regarding pathology outcomes, out of the total number of lesions, 32 (58%) were benign, 18 (33%) were identified as invasive ductal carcinoma (IDC), and 3 (5%) were classified as ductal carcinoma in situ (DCIS). Among the 34 lesions that are neither IDC nor DCIS, 2 were identified as malignant. The ROC analysis of the logistic regression model (Figure 4), based on 55 observations, demonstrates an AUC of 0.90 with a standard error of 0.04. The 95% confidence interval for the AUC ranges from 0.78 to 0.96.

#### 3.2.2. QUS Parameters According to the Final Diagnosis and Pathologic Outcome

The actual representation of the QUS parameters is illustrated in Figure 5. The median AC value for benign lesions is 0.506 with an IQR of 0.402–0.623, while for malignant lesions, the median value is 0.666 with an IQR of 0.609–0.731 and a statistically significant *p*-value of <0.001 (Table 4).

For the SoS parameter, the median value for benign lesions is 1542 (IQR: 1525–1558.5), and for malignant lesions, it is 1565 (IQR: 1550–1579), with a significant *p*-value of 0.002. The ESD has a median value of 91.53 (IQR: 86.46–97.71) for benign lesions and 83.76 (IQR: 74–90.6) for malignant lesions, with a *p*-value of 0.001. However, the ESC shows no significant difference between benign and malignant lesions, with median values of 2.396 (IQR: 1.926–3.008) and 2.6 (IQR: 1.987–3.733), respectively, and a *p*-value of 0.16.

Additionally, the data provides a comparison of QUS parameters between IDC (*n* = 18, 86%) and DCIS (*n* = 3, 14%) lesions. The AC, SoS, ESD, and ESC values for both IDC and DCIS lesions are reported with their respective median values and IQRs. Due to small sample sizes, *p*-values were not calculated in this smaller subgroup analysis.

### 3.3. Quantitative Ultrasound Parameters Help Identify Unclear Tumor Types in Traditional B-Mode Images

Figure 6 presents QUS image reconstructions of the cases where the tumor type is hardly characterized through the conventional B-mode image. The cyst (Figure 6a) is mostly composed of water, hence showing a significantly low attenuation coefficient (0.1570 dB/cm/MHz) compared to the malignant tumor (Figure 6c) (1.012 dB/cm/MHz). Speed of sound is another biomarker for differentiating benign tumors and malignant lesions. The malignant lesion demonstrates a higher speed-of-sound value (1571 m/s) compared to the benign tumor (Figure 6b) (1551 m/s). The malignant tumor (Figure 6c) shows a lower scatter diameter and higher scatter concentration than the benign tumors (Figure 6b). The malignant tumor shows a lower scatter diameter and higher scatter concentration than one of the benign tumors. In Figure 6c, the position and actual geometrical shape of the malignant lesion are rarely possible to be identified with the B-mode image. However, the QUS images, especially the attenuation coefficient and speed-of-sound images, visualize where the abnormal biomechanical properties are shown and make it possible to delineate lesion location with precision. 

For BI-RADS categories 3 and 4, the QUS parameters for the five benign lesions and one malignant lesion are provided (Table 5). For BI-RADS category 3, the AC values range from 0.39 to 0.560, SoS values range from 1492 to 1590, ESD values range from 64.94 to 103, and ESC values range from 1.293 to 2.847. For BI-RADS category 4, the QUS parameters for the three lesions are listed. The AC values range from 0.427 to 0.52, SoS values range from 1507 to 1538, ESD values range from 55.6 to 94.12, and ESC values range from 1.58 to 4.153.

## 4. Discussion

In this initial investigation, our objective was to assess the effectiveness of a quantitative ultrasound (QUS) algorithm that utilizes artificial intelligence for the reconstruction of attenuation coefficient (AC), speed of sound (SoS), effective scatterer diameter (ESD), and effective scatterer concentration (ESC) in breast ultrasound radiofrequency (RF) signals. We analyzed the QUS parameters in comparison with biopsy results to evaluate the algorithm’s ability to differentiate between benign and malignant breast lesions. Our results indicate that the inclusion of QUS parameters alongside conventional B-mode ultrasound has the potential to improve diagnostic accuracy, particularly for lesions that present challenges with unclear characteristics.

The results demonstrated that the QUS parameters AC, SoS, and ESD showed significant differences between benign and malignant lesions. The median values of AC and SoS were higher in malignant lesions than in benign lesions, while the ESD median value was lower in malignant lesions. These findings align with previous studies reporting that malignant breast lesions generally exhibit higher attenuation coefficients and speeds of sound, along with smaller scatterer sizes [22]. Quantitative ultrasound techniques, such as QUS, have been increasingly utilized in breast cancer detection due to their ability to provide objective and reproducible measurements of tissue properties [23,24]. However, ESC showed no significant difference between benign and malignant lesions, which is consistent with other studies showing that ESC has limited discriminatory power for distinguishing between benign and malignant lesions [25,26]. Some researchers have also suggested that the combination of multiple QUS parameters may improve the diagnostic performance of breast ultrasound [27,28]. We also observed that the QUS parameters provided valuable information in the intermediate BI-RADS categories 3 and 4, which are often challenging to classify based on conventional B-mode ultrasound alone. The quantitative nature of QUS parameters may help improve the diagnostic accuracy of breast ultrasound, especially in lesions with ambiguous findings [29,30].

A logistic regression model was developed using the QUS parameters, which achieved an AUC of 0.8995, indicating a high diagnostic performance for discriminating between benign and malignant breast lesions. This result suggests that the proposed QUS algorithm has the potential to enhance the diagnostic accuracy of breast ultrasound and may serve as a valuable adjunct to conventional B-mode imaging for breast cancer detection and characterization [31].

There are some limitations to our study. Firstly, the sample size was relatively small, and the majority of malignant lesions were IDC. Further studies with larger and more diverse patient populations are needed to confirm the generalizability of our findings. Secondly, the algorithm’s performance in specific subgroups, such as women with dense breasts or those at high risk for breast cancer, was not evaluated in this study. Future research could explore the potential expansion of the algorithm’s applications, such as in monitoring treatment response, predicting prognosis, or evaluating other breast-related conditions.

## 5. Conclusions

Our pilot study preliminarily indicates that integrating AI with quantitative ultrasound analysis, specifically through reconstruction of attenuation coefficient, speed of sound, effective scatterer diameter, and effective scatterer concentration, shows promise for enhancing breast cancer detection and characterization accuracy. This combined approach could potentially improve diagnostics, especially for lesions with unclear B-mode ultrasound findings. However, the impact of the case distribution on our findings underscores the need for further research with a more balanced representation of benign and malignant cases to confirm these initial results.

## Figures and Tables

**Figure 1 diagnostics-14-00419-f001:**
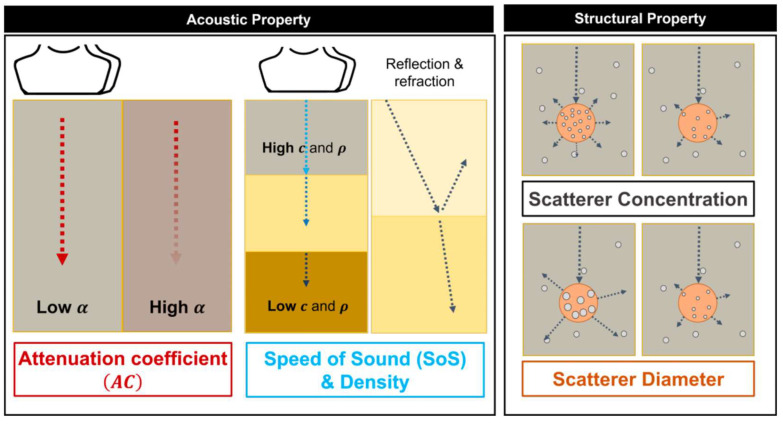
Biomechanical properties considered in the proposed quantitative imaging system.

**Figure 2 diagnostics-14-00419-f002:**
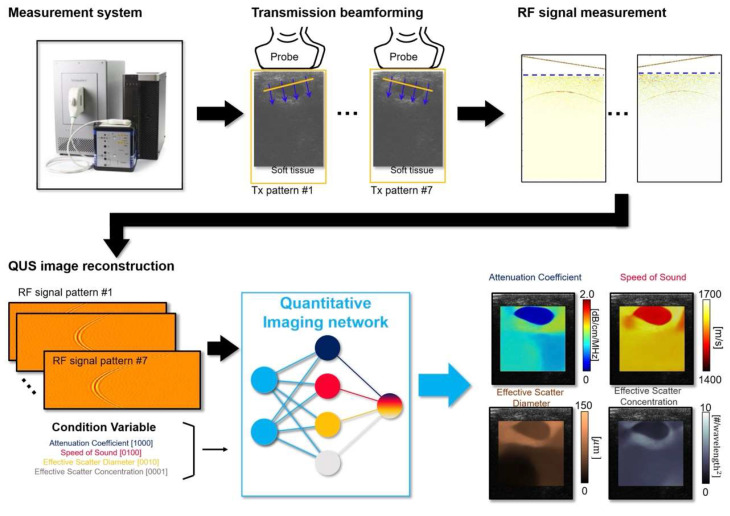
Workflow of the proposed quantitative imaging system.

**Figure 3 diagnostics-14-00419-f003:**
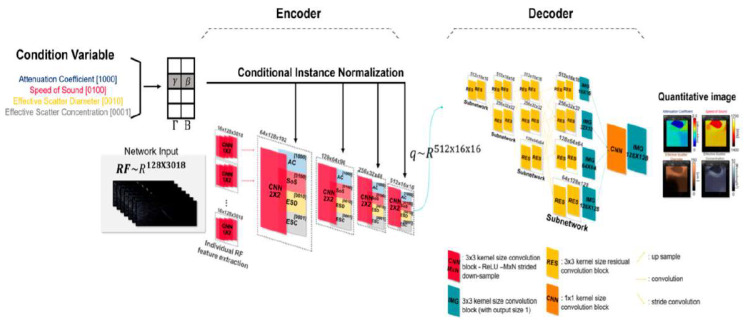
Details of the neural network configuration.

**Figure 4 diagnostics-14-00419-f004:**
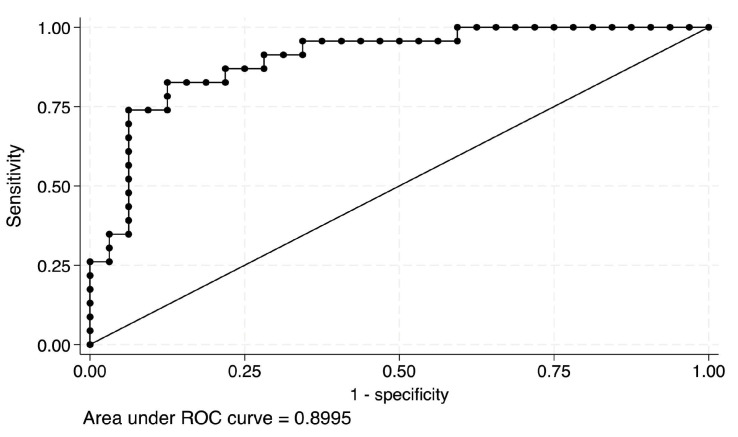
The Receiver Operating Characteristic Curve analysis of the logistic regression model.

**Figure 5 diagnostics-14-00419-f005:**
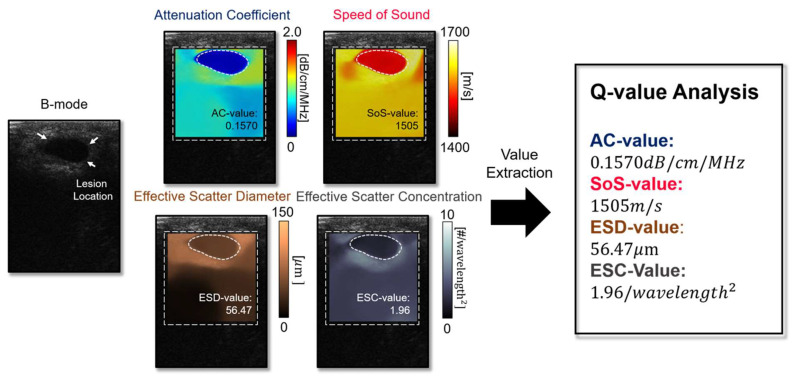
The actual representation of the QUS parameters.

**Figure 6 diagnostics-14-00419-f006:**
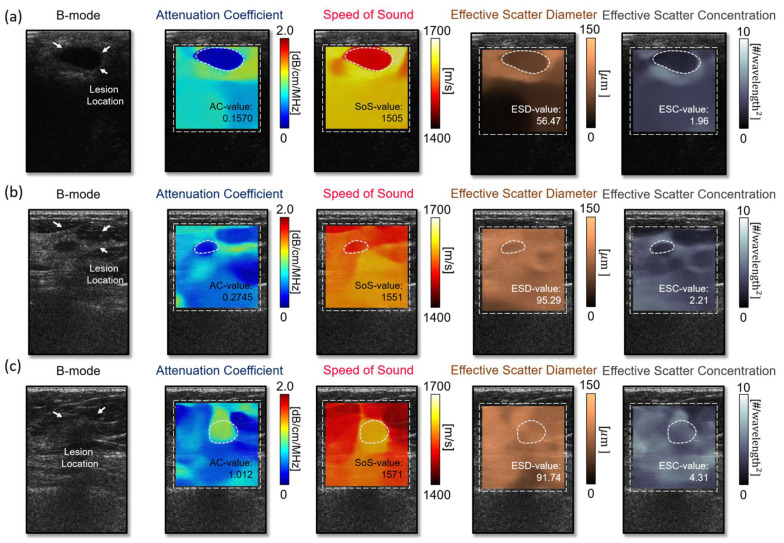
Quantitative ultrasound parameter extraction. (**a**) cyst, (**b**) benign tumor and (**c**) malignant tumor.

**Table 1 diagnostics-14-00419-t001:** The optimization procedure of the proposed quantitative imaging neural network.

Algorithm Optimization
**Input:** $\left(RF,\;cond,\;I_{cond}\;\right)\sim \;{Dataset}_{\;}$
**Initialize:** neural network parameter: θ
1: **For** it **in** iterations **do**
2: Split $Dataset_{train}\;and\;Dataset_{test}\leftarrow Dataset\;$
3: **Composed train batch:** L$\left(RF_{tr},\;cond_{tr},\;I_{cond_{Tr}}\;\right)\sim \;Dataset_{train}$
4: **Calculate loss:** $Meta\;DSA\;loss\;{L_{QI}}_{\;}^{\;}(RF_{tr},\;cond_{tr},\;I_{cond_{Tr}})$
5: **Model Optimization** θ = $\theta -\lambda \frac{\partial \left({L_{QI}}_{\;}^{\;}(RF_{tr},\;cond_{tr},\;I_{cond_{Tr}})\right)}{\partial \theta }$
6: **end for**

**Table 2 diagnostics-14-00419-t002:** Logistic regression of QUS parameters for differential diagnosis between benign and malignancy.

QUS Parameter	Odds Ratio	Standard Error	z	*p*-Value	95% CI
AC	41,734.12	144,309.9	3.08	0.002	47.55–3.66 × 10^7^
SoS	1.02	0.016	1.33	0.183	0.99–1.05
ESD	0.91	0.032	−2.70	0.007	0.85–0.97
ESC	1.08	0.48	0.18	0.86	0.46–2.57

AC = Attenuation Coefficient (dB/cm/MHz), SoS = Speed of Sound (m/s), ESD = Effective Scatterer Diameter (µm), ESC = Effective Scatterer Concentration (#/wavelength^2).

**Table 3 diagnostics-14-00419-t003:** Patient and Lesion Characteristics (*N* = 55).

	Benign, 32 (58)	Malignant, 23 (42)	Total, 55 (100)
Age (mean ± SD)	41.9 (1.8)	53.8 (2.3)	
Tumor size (mm), median (IQR)	14 (10–17)	20.5 (14–28)	
BI-RADS category, *n* (%)			
1	1 (3)	0 (0)	1 (3)
2	5 (14)	0 (0)	5 (14)
3	5 (14)	1 (3)	6 (17)
4	2 (6)	1 (3)	3 (8)
5	0 (0)	1 (3)	1 (3)
6	0 (0)	20 (64)	20 (64)
Pathology outcome, *n* (%)			
IDC	0 (0)	18 (33)	18 (33)
DCIS	0 (0)	3 (5)	3 (5)
N/A	32 (58)	2 (4)	34 (6)

BI-RADS = Breast Imaging-Reporting and Data System, SD = standard deviation, IQR = interquartile range, IDC = Invasive Ductal Carcinoma, DCIS = Ductal Carcinoma In Situ, AC = Attenuation Coefficient, SoS = Speed of Sound, ESD = Effective Scatterer Diameter, ESC = Effective Scatterer Concentration, N/A=Not Applicable.

**Table 4 diagnostics-14-00419-t004:** Measurement of quantitative ultrasound parameters according to the final diagnosis and pathologic outcome.

QUS Parameters	Benign, 32 (58)	Malignant, 23 (42)	*p*-Value
AC, median (IQR)	0.506 (0.402–0.623)	0.666 (0.609–0.731)	<0.001
SoS, median (IQR)	1542 (1525–1558.5)	1565 (1550–1579)	0.002
ESD, median (IQR)	91.53 (86.46–97.71)	83.76 (74–90.6)	0.001
ESC, median (IQR)	2.396 (1.926–3.008)	2.6 (1.987–3.733)	0.16
	IDC, 18 (86)	DCIS, 3 (14)	
AC, median (IQR)	0.664 (0.616–0.73)	0.682 (0.577–0.877)	… *
SoS, median (IQR)	1565 (1552–1577)	1577 (1525–1579)	…
ESD, median (IQR)	83.2 (74–88.8)	97 (90.33–101.9)	…
ESC, median (IQR)	2.684 (2.22–3.804)	3.148 (2.525–3.733)	…

* Due to small sample sizes, *p*-values were not calculated in this smaller subgroup analysis. AC, Attenuation Coefficient, quantifies the loss of acoustic energy during the propagation of ultrasound waves, and it is measured in dB/cm/MHz; SoS, Speed of Sound, refers to the ultrasound propagation speed in tissue, which is measured in m/s; ESD, Effective Scatterer Diameter quantifies the diameter of tissue microstructures and is measured in µm; ESC, Effective Scatterer Concentration refers to the number of scattering sources in the unit area and is measured in #/wavelength^2. IDC = Invasive Ductal Carcinoma, DCIS = Ductal Carcinoma In Situ.

**Table 5 diagnostics-14-00419-t005:** Quantitative ultrasound parameters in Bi-RADS categories 3 and 4.

BI-RADS Category	QUS Parameter	Benign, 7					Malignant, 2
		Lesion 1	Lesion 2	Lesion 3	Lesion 4	Lesion 5	Lesion 6
3	AC	0.39	0.507	0.560	0.461	0.505	0.8061
	SoS	1508	1542	1492	1530	1557	1590
	ESD	86.67	103	77.53	64.94	96.12	89.88
	ESC	2.4	1.293	2.392	2.847	2.118	1.976
		Lesion 7	Lesion 8				Lesion 9
4	AC	0.52	0.427				0.432
	SoS	1538	1507				1525
	ESD	94	94.12				55.6
	ESC	4.153	3.247				1.58

Comparing the values of AC, SoS, ESD, and ESC, a perfect distinction is difficult to achieve, but some differences do exist. For example, malignant lesions tend to have higher AC values and lower ESD values. However, these patterns do not apply to all lesions, making it difficult to perfectly differentiate them using these values alone. AC = Attenuation Coefficient (dB/cm/MHz), SoS = Speed of Sound (m/s), ESD = Effective Scatterer Diameter (µm), ESC = Effective Scatterer Concentration (#/wavelength^2).

## Data Availability

The raw data supporting the conclusions of this article will be made available by the authors on request.

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
