# Peer review of "Artificial Intelligence-Enhanced Quantitative Ultrasound for Breast Cancer: Pilot Study on Quantitative Parameters and Biopsy Outcomes"

_diagnostics, 2024, doi:10.3390/diagnostics14040419_

Round 1

Reviewer 1 Report

Comments and Suggestions for Authors

1. Page 2, rows 65-74 - should be deleted from " The introduction should..." as they are recommendations for authors

2. Page 3, row 104 - refraction instead of retraction

3. Page 3, row 109 - SoS instead of SOS

4. Page 4, row 130 - comma not dot after featurs, generates instead of generated, conditional instead of Conditional

5. Authors are kindly asked to explain the use of a 5MHz transducer, taking into consideration that breast ultrasound is mainly performed with 8-9 MHz transducers.

6. Authors are kindly asked to explain the role of fine needle aspiration biopsy, as they say that pathology report is considered ground truth and they differentiate pathologies between IDC, DCIS and benign lesions.

7. The number of 55 cases, from which 32 are benign and 23 are malignant (and from which 20 are BI-RADS 6 - so already known as malignant) can induce statistical missaproaches. So I think that the conclusions might be reformulated taking into consideration this aspect.

8. From 31 refferences only 3 are from 2019 to present (the 4th being a conference in 2021). Maybe the authors could find some new articles in the field of interest.

Comments on the Quality of English Language

no comments

Author Response

We appreciate the constructive feedback and have made the following revisions based on your recommendations:

  1. Page 2, rows 65-74 - should be deleted from " The introduction should..." as they are recommendations for authors

Response

We have removed the specified section from the introduction that contained recommendations for authors, ensuring the focus remains solely on our research findings and their implications.

  1. Page 3, row 104 - refraction instead of retraction

Response

Corrections have been made to accurately reflect the term "refraction" in place of "retraction," aligning with the proper terminology in the context of ultrasound physics.

  1. Page 3, row 109 - SoS instead of SOS

Response

The abbreviation for Speed of Sound has been standardized to "SoS" to maintain consistency.

  1. Page 4, row 130 - comma not dot after featurs, generates instead of generated, conditional instead of Conditional

Response

Grammatical adjustments have been made, including the correction of a punctuation error, verb tense, and the use of "conditional".

  1. Authors are kindly asked to explain the use of a 5MHz transducer, taking into consideration that breast ultrasound is mainly performed with 8-9 MHz transducers.

Response

The rationale for using a 5MHz transducer, despite the common use of higher frequency transducers in breast ultrasound, has been elaborated upon in the methods section. This addition provides a comprehensive explanation for our choice, highlighting the benefits of deeper tissue penetration and broader tissue characterization capabilities.

“The choice to use a 5MHz transducer for the study, despite breast ultrasound typically being performed with higher frequency transducers (8-9 MHz), can be attributed to several factors. Lower frequency transducers, like the 5MHz, have deeper penetration capabilities, which can be crucial for visualizing deeper breast tissues and lesions. This can be especially beneficial in patients with denser breast tissues where higher fre-quencies may not penetrate as effectively. Moreover, the 5MHz transducer can still provide sufficient resolution for the quantitative analysis required in this study, while also allowing for a broader range of tissue characterization due to its wavelength properties. These characteristics make the 5MHz transducer a suitable choice for the specific re-quirements and objectives of the pilot study on AI-enhanced quantitative ultrasound for breast cancer diagnosis.”

  1. Authors are kindly asked to explain the role of fine needle aspiration biopsy, as they say that pathology report is considered ground truth and they differentiate pathologies between IDC, DCIS and benign lesions.

Response

The role of Fine Needle Aspiration Biopsy in our study's diagnostic process has been further clarified in the methods section, emphasizing its importance in differentiating between various breast pathologies.

“Fine Needle Aspiration Biopsy plays a critical role in breast cancer diagnosis as it in-volves the extraction of cells from breast lesions for microscopic examination. FNAB serves as a minimally invasive method to obtain tissue samples, which are then analyzed to differentiate between various pathologies, including Invasive Ductal Carcinoma (IDC), Ductal Carcinoma In Situ (DCIS), and benign lesions.”

  1. The number of 55 cases, from which 32 are benign and 23 are malignant (and from which 20 are BI-RADS 6 - so already known as malignant) can induce statistical missaproaches. So I think that the conclusions might be reformulated taking into consideration this aspect.

Response

In response to the distribution of benign and malignant cases in our study, we have revised our conclusion to more accurately reflect the preliminary nature of our findings and the need for further research with a more balanced case distribution.

In conclusion, our pilot study demonstrated that a quantitative ultrasound algorithm using artificial intelligence reconstruction of attenuation coefficient, speed of sound, effective scatterer diameter, and effective scatterer concentration in breast ultrasound radiofrequency signals has the potential to improve the diagnostic accuracy of breast cancer detection and characterization. The integration of quantitative ultrasound pa-rameters with conventional B-mode ultrasound may enhance the overall diagnostic performance, particularly in challenging lesions with ambiguous findings.

Is changed to

Our study preliminarily indicates that integrating AI with quantitative ultrasound analysis, specifically through reconstruction of attenuation coefficient, speed of sound, effective scatterer diameter, and effective scatterer concentration, shows promise for enhancing breast cancer detection and characterization accuracy. This combined approach could potentially improve diagnostics, especially for lesions with unclear B-mode ultrasound findings. However, the impact of the case distribution on our findings underscores the need for further research with a more balanced representation of benign and malignant cases to confirm these initial results.

  1. From 31 refferences only 3 are from 2019 to present (the 4th being a conference in 2021). Maybe the authors could find some new articles in the field of interest.

Response

We have updated our references to include more recent studies, ensuring our research is grounded in the latest advancements and discussions within the field.

  1. Quiaoit, K.; DiCenzo, D.; Fatima, K.; Bhardwaj, D.; Sannachi, L.; Gangeh, M.; Sadeghi-Naini, A.; Dasgupta, A.; Kolios, M.C.; Trudeau, M. Quantitative ultrasound radiomics for therapy response monitoring in patients with locally advanced breast cancer: Multi-institutional study results. PloS one 2020, 15, e0236182.
  2. Yuan, W.-H.; Hsu, H.-C.; Chen, Y.-Y.; Wu, C.-H. Supplemental breast cancer-screening ultrasonography in women with dense breasts: a systematic review and meta-analysis. British journal of cancer 2020, 123, 673-688.

Your feedback has been invaluable in enhancing the accuracy, relevance, and clarity of our manuscript.

Reviewer 2 Report

Comments and Suggestions for Authors

This study presents deep learning based quantitative ultrasound parameter estimation methods and their applications in classification of benign and malignant breast tumors. This is interesting and presents some novelty. However, the writing should be improved. Please see the following concerns.

1.        Line 19. Please define ‘RF.’

2.        Lines 65-74. What are the purposes of these sentences? These seem unrelated to the manuscript and should be removed.

3.        Lines 96-98. “… reconstruct the tissue AC, SoS, ESD, and ESC of the tissue, by analyzing amplitude, phase, and envelop statistics of the radio-frequency signal, respectively.” Is any of the four parameters related to envelop statistics? Conventional methods for estimating the four parameters are generally performed in the frequency-domain.

4.        Line 115 and similar occurrences. ‘scatter’ should be ‘scatterer.’

5.        Line 128. ‘the RF’ should be ‘the RF signal(s).’

6.        Lines 129-130. ‘features. q’ to ‘feature q.’ And, q is suggested to be bold, because it is a feature vector.

7.        Figures 2 and 3. Please improve the figure resolution.

8.        Line 141 and similar occurrences. ‘k-wave’ to ‘k-Wave.’

9.        Line 144. ‘RF’ has been defined in the Introduction section. No need to redefine it here.

10.     Line 185 and similar occurrences. ‘b-mode’ to ‘B-mode.’

11.     Line 191 and similar occurrences. ‘radio-frequency’ to ‘RF.’

12.     Line 196. ‘1e^(−4)’ to ’10^(−4).’

13.     Please provide a figure illustrating the training and validation loss as a function of training epochs.

14.     Line 341 and similar occurrences. ‘p’ should be italic.

15.     Figure 5. In the right part of this figure, the bottom line ‘ESD-value’ should be ‘ESC-value.’

16.     Figure 6. Please improve the resolution.

17.     Please note that the above concerns are not meant to be exhaustive. It is the authors’ responsibility to carefully proofread the manuscript.

18.     Please note that the above concerns are not meant to be exhaustive. It is the authors’ responsibility to carefully proofread the manuscript.

Comments on the Quality of English Language

Moderate improvement needed.

Author Response

Reviewer 2.

Thank you very much for taking the time to review this manuscript. Please find the detailed responses below and the corresponding revisions highlighted the re-submitted files

  1. Line 19. Please define ‘RF.’

Response

Thank you for pointing this out, we updated text in the manuscript.

  1. Lines 65-74. What are the purposes of these sentences? These seem unrelated to the manuscript and should be removed.

Response

Thank you for pointing this out, we updated text in the manuscript.

  1. Lines 96-98. “… reconstruct the tissue AC, SoS, ESD, and ESC of the tissue, by analyzing amplitude, phase, and envelop statistics of the radio-frequency signal, respectively.” Is any of the four parameters related to envelop statistics? Conventional methods for estimating the four parameters are generally performed in the frequency-domain.

Response

AC (Attenuation Coefficient), SoS (Speed of Sound), ESD (Effective Scatterer Diameter), and ESC (Effective Scatterer Concentration) by analyzing amplitude, phase, and envelope statistics of the radio-frequency signal, and these parameters are estimated through methods that analyze the frequency domain characteristics of the ultrasound signal.

We added intext,

These parameters are estimated through methods that analyze the frequency domain characteristics of the ultrasound signal.

  1. Line 115 and similar occurrences. ‘scatter’ should be ‘scatterer.’

Response

Thank you for pointing this out, we updated text in the manuscript. Corrected to 'scatterer' for accurate terminology.

  1. Line 128. ‘the RF’ should be ‘the RF signal(s).’

Response

Thank you for pointing this out, we updated text in the manuscript.

  1. Lines 129-130. ‘features. q’ to ‘feature q.’ And, q is suggested to be bold, because it is a feature vector.

Response

fixed

Thank you for pointing this out, we updated text in the manuscript

  1. Figures 2 and 3. Please improve the figure resolution.

Response

The resolution of Figures 2 and 3 has been enhanced for better visualization.

  1. Line 141 and similar occurrences. ‘k-wave’ to ‘k-Wave.’

Response

Thank you for pointing this out, we updated text in the manuscript. Corrected to match proper naming conventions.

  1. Line 144. ‘RF’ has been defined in the Introduction section. No need to redefine it here.

Response

Thank you for pointing this out, we updated text in the manuscript.

  1. Line 185 and similar occurrences. ‘b-mode’ to ‘B-mode.

Response

Thank you for pointing this out, we updated text in the manuscript.

  1. Line 191 and similar occurrences. ‘radio-frequency’ to ‘RF.’

Response

Thank you for pointing this out, we updated text in the manuscript.

  1. Line 196. ‘1e^(−4)’ to ’10^(−4).’

Response

Thank you for pointing this out, we updated text in the manuscript.

  1. Please provide a figure illustrating the training and validation loss as a function of training epochs. 

Response

We added as a supplementary file and the sentence.

“A figure illustrating the training and validation loss as a function of training epochs is added as a supplementary file.”

  1. Line 341 and similar occurrences. ‘p’ should be italic.

Response

Thank you for pointing this out, we updated text in the manuscript.

  1. Figure 5. In the right part of this figure, the bottom line ‘ESD-value’ should be ‘ESC-value.’

Response

Thank you for pointing this out, we corrected the label in Figure 5 for accuracy.

  1. Figure 6. Please improve the resolution.

 Response

Thank you for pointing this out, we enhanced the resolution of Figure 6 for better clarity.

  1. Please note that the above concerns are not meant to be exhaustive. It is the authors’ responsibility to carefully proofread the manuscript.

Response

Certainly, we understand and acknowledge the statement that the above concerns are not exhaustive, and it is the authors' responsibility to carefully proofread the manuscript. The authors will take this responsibility seriously and conduct a comprehensive proofreading to ensure the highest quality and accuracy of the manuscript.

  1. Please note that the above concerns are not meant to be exhaustive. It is the authors’ responsibility to carefully proofread the manuscript.

Response

Certainly, we understand and acknowledge the statement that the above concerns are not exhaustive, and it is the authors' responsibility to carefully proofread the manuscript. The authors will take this responsibility seriously and conduct a comprehensive proofreading to ensure the highest quality and accuracy of the manuscript.

Round 2

Reviewer 2 Report

Comments and Suggestions for Authors

Thanks for the revision. It has addressed my concerns.